biochemistry/bioengineering/environmental science

*Aspergillus*, ligninolytic enzymes, azo dyes, degradation, detoxification

**Author for correspondence:**
Xiaolin Xu
e-mail: xxl_food@shzu.edu.cn

This article has been edited by the Royal Society of Chemistry, including the commissioning, peer review process and editorial aspects up to the point of acceptance.

# Degradation and detoxification of azo dyes with recombinant ligninolytic enzymes from *Aspergillus* sp. with secretory overexpression in *Pichia pastoris*

Siqi Liu, Xiaolin Xu, Yanshun Kang, Yingtian Xiao and Huan Liu

Key Laboratory for Green Processing of Chemical Engineering of Xinjiang Bingtuan/School of Chemistry and Chemical Engineering, Shihezi University, Shihezi 832003, People's Republic of China

XX, 0000-0002-1741-0167

Ligninolytic enzymes, including laccase (Lac), manganese peroxidase (MnP) and lignin peroxidase (LiP), have attracted much attention in the degradation of contaminants. Genes of Lac (1827 bp), MnP (1134 bp) and LiP (1119 bp) were cloned from *Aspergillus* sp. TS-A, and the recombinant Lac (69 kDa), MnP (45 kDa) and LiP (35 kDa) were secretory expressed in *Pichia pastoris* GS115, with enzyme activities of 34, 135.12 and 103.13 U l$^{-1}$, respectively. Dyes of different structures were treated via the recombinant ligninolytic enzymes under the optimal degradation conditions, and the result showed that the decolourization rate of Lac on Congo red (CR) in 5 s was 45.5%. Fourier-transform infrared spectroscopy, gas chromatography–mass spectrometry analysis and toxicity tests further proved that the ligninolytic enzymes could destroy the dyes, both those with one or more azo bonds, and the degradation products were non-toxic. Moreover, the combined ligninolytic enzymes could degrade CR more completely compared with the individual enzyme. Remarkably, besides azo dyes, ligninolytic enzymes could also degrade triphenylmethane and anthracene dyes. This suggests that ligninolytic enzymes from *Aspergillus* sp. TS-A have the potential for application in the treatment of contaminants.

# 1. Introduction

Synthetic dyes are widely used in the paper, textile, pharmaceutical and food industries, and thus, a significant amount of dye containing effluent from these industries is discharged into the environment. [1,2]. Most of these dyes are highly toxic, mutagenic and carcinogenic [2–4]. Among the synthetic dyes, azo dyes, the largest and most widely used dyes, are characterized by one or more azo bonds (–N = N–) connected to aromatic rings [2,5], and are hard to destroy [2]. Moreover, the process of dye pollutant removal is typically associated with cost, speed and detoxification.

Multiple technologies have been introduced to remove azo dyes such as sorption, chemical oxidation, electrochemical degradation and ultrafiltration [4–6], but these methods are limited; for example, the physical processes do not destroy the dyes, the chemical processes are non-selective, operational cost is high and toxic by-products are produced [2]. Owing to cost effectiveness and environmental benignity, biodecolourization has been recognized as a prospective treatment option for dye removal [7–9]. In recent years, among all microbes, fungi have received the most attention for the treatment of dye wastewater, owing to their strong ability to degrade complex organic compounds completely. In addition to white-rot fungi [10], *Aspergillus* [11,12] has been used to remove dyes through absorption by mycelium, degradation with ligninolytic enzymes or a combination of both. However, the long fungal growth processes and the complex catalytic systems limit the application of biological treatment. Therefore, the enzymatic treatment has more valuable advantages than the microbial treatment in wastewater remediation.

Ligninolytic enzymes used for the degradation of dyes mainly include laccase (Lac; EC 1.10.3.2), manganese peroxidase (MnP; EC 1.11.1.13) and lignin peroxidase (LiP; EC 1.11.1.14) [13]; their contributions may vary for different fungi. Lac, which has a high redox potential and is composed of connected cupredoxin-like domains twisted into a tight globule [14,15], can catalyse the ring cleavage reaction of aromatic compounds [16,17]. Lacs from different sources have shown various catalytic properties. Legerská reported decolourization rates of 72.8% and 45.3% for Orange 2 and acid Orange 6 azo dyes, respectively, using Lac from *Trametes versicolor* [18]. Iark found that the decolourization rates of Congo red (CR) degraded by Lac from *Oudemansiella canarii* reached 80% [19]. MnP has been used to oxidize Mn(II) to Mn(III) to degrade some recalcitrant organic pollutants such as dyes and phenol [20]. MnPs exist as different types of isozymes [21], and their diverse amino acid sequences vary in terms of the C-terminal and the number of lysine residues [22]. Zhang *et al.* [23] reported that both dyes and polycyclic aromatic hydrocarbons were degraded by MnP from *Trametes* sp. 48424. Fungal LiPs have been reported to catalyse the $H_2O_2$-dependent oxidative depolymerization of lignin [24]; these LiPs are globular and mostly helical glycoproteins of about 40 kDa, ranging from 343 to 344 amino acids [25]. However, the enzymatic treatment of dyes is mainly limited by one key factor; insufficient enzyme production.

To improve the production of enzymes, ligninolytic enzymes from different fungal resources have been heterologously expressed in *Escherichia coli* [26] and *Pichia pastoris* [27]. For example, in one study, the Lac gene from *Trametes trogii* was expressed in *P. pastoris*, and thermostable recombinant Lac with a half-life of 45 min at 70°C was found capable of decolourizing azo dyes, such as acid red 26 [28]. Fan *et al.* [29] reported that a Lac gene from *Trametes* sp. 48424 was expressed in *P. pastoris*; a high yield of the recombinant Lac was obtained and the enzyme could decolourize different dyes.

Nevertheless, the enzymatic treatment of dyes also has disadvantages; for example, the enzymatic degradation of azo dyes leads to the formation of toxic products, mainly amines. It is important to identify and evaluate degradation products. Parts of the degradation intermediates have been reported to have low molecular weight. Azo dyes degraded by ligninolytic enzymes from *Aspergillus niger* have been found to yield products that contain structures with a small number of benzene rings, such as sodium naphthalene sulfonate and cycloheptadienylium; moreover, the toxicity of the products was significantly decreased compared with that of the original dye [12]. Iark *et al.* reported the degradation of CR by Lac with an *m/z* value of 255.23 ($C_8H_3N_2O_8^-$), which is an oxygenated compound with an open benzene ring and reduced toxicity [19]. Similar metabolites were obtained in another study from the degradation of CR with Lac from *Ganoderma lucidum* [30]. These results suggest that azo dyes can be degraded and detoxified by ligninolytic enzymes. Moreover, the precise mechanisms of the degradation of these chromophore groups have been determined [12,30]. In a previous study, *Aspergillus* sp. TS-A CGMCC12964 was shown to robustly degrade an azo dye with ligninolytic enzymes [11], although the degradation products and pathway were not clear.

In the current study, ligninolytic enzymes genes were cloned from *Aspergillus* sp. TS-A CGMCC 12964 [11] which has been reported to effectively remove azo dye, and recombinant enzymes were secretory

**Table 1.** Chemical structure and maximum absorbance of dyes.

| dye | type | maximum wavelength of absorbance (nm) | chemical structure |
|-----|------|----------------------------------------|---------------------|
| mordant yellow 1 | azo | 362 |  |
| Congo red | azo | 497 |  |
| disperse blue 2BLN | anthracene | 560 |  |
| bromophenol blue | triphenylmethane | 422 |  |

expressed in *P. pastoris* GS115. The catalysis reaction of the ligninolytic enzymes was explored by the decolourization of different dyes. The degradation products of azo dye were determined using Fourier-transform infrared (FTIR) and gas chromatography-mass spectrometry (GC-MS), and the possible degradation pathways and detoxification ability were estimated.

# 2. Material and methods

## 2.1. Dye and chemicals

Most synthetic dyes are toxic and difficult to degrade, causing environment pollution [31]. The dyes mordant yellow 1 (MY1; CAS: 584-42-9), CR (CAS: 573-58-0), disperse blue 2BLN (CAS: 12217-79-7) and bromophenol blue (CAS: 115-39-9) (table 1) are commonly used in the textile industry. The dyes used in this study were purchased from Aladdin Reagent Co. Ltd. (Shanghai, China). 2, 2′-azion-bis (3-ethyl-benzothiazoline)-6-sulfonic acid (ABTS) was purchased from BIO BASIC Inc. (Canada). All the chemicals used in this work were of analytical grade purity or above.

## 2.2. Strains, plasmids and media

*Aspergillus* sp. TS-A CGMCC 12964, a previously screened strain, was obtained from the Key Laboratory for Green Processing of Chemical Engineering of Xinjiang Bingtuan, School of Chemistry and Chemical Engineering, Shihezi University. The strain was maintained in a Czapek-Dox medium. *Pichia pastoris* GS115 was purchased from Invitrogen. Plasmid pPIC9 K was stored in a laboratory. The following media were prepared: Luria Bertani, yeast extract peptone dextrose (YPD), minimal dextrose (MD), minimal methanol (MM), buffered glycerol complex (BMGY) and buffered methanol complex (BMMY).

## 2.3. Cloning and expression of MnP, LiP and Lac

*Aspergillus* sp. TS-A was cultivated in Czapek–Dox medium. After 48 h, the mycelium was collected by filtration and then ground. DNA fragments were obtained using a fungal gDNA extraction kit. The DNA fragments of MnP, LiP and Lac were digested with *Eco*RI and *Not*I and inserted into the same sites of the pPIC9 K expression vector, resulting in the recombinant plasmids pPIC9 K-Lac, pPIC9 K-MnP and pPIC9 K-LiP. The recombinant plasmids were linearized by *Sal*I digestion and then electroporated into competent *P. pastoris* GS115; afterwards, they were plated onto MD and MM plates to select the transformants. The positive transformants were inoculated in the YPD medium and then a single colony was selected. Polymerase chain reaction (PCR) was used to analyse recombinant target genes to verify whether the transformant was successfully transformed.

## 2.4. Determination of enzyme activities and protein

The recombinant yeasts GS115/Lac, GS115/MnP and GS115/LiP were inoculated into a shake flask of 100 ml BMGY media and incubated under 30°C and at 200 rpm shake speed. Afterward, recombinant yeasts were precipitated by centrifugation for 24 h at 4°C and resuspended in 100 ml of BMMY media. Then the cultures were grown at the same condition for 96 h, with 0.5% (v/v) methanol addition daily. Secreted enzyme activities in cultures were measured daily. The supernatants were used as the recombinant enzymes, collected by centrifugation and filtration through 0.22 μm.

The activities of recombinant Lac, MnP and LiP were assayed spectrophotometrically in a cell-free extract. ABTS, $MnSO_4$ and veratryl alcohol were used as substrates. One unit of enzymatic activity was defined as the amount of enzyme transforming 1 μmol of the substrate per minute. The determined enzyme activities were referenced to the results of Pan *et al.* [32]. Protein concentrations were measured using the Bradford assay (Coomassie brilliant blue) and bovine serum albumin as the standard. Recombinant enzymes were verified using sodium dodecyl sulfate–polyacrylamide-gel electrophoresis (SDS-PAGE) where the stacking and separating gels contained 5% and 12% polyacrylamide, respectively.

## 2.5. Characterization of recombinant enzymes

### 2.5.1. Effect of pH and temperature on enzyme activity

To evaluate the optimal pH for recombinant enzymes using ABTS, $MnSO_4$ and veratryl alcohol as substrates, the enzymatic reaction was performed in buffers of different pH values (1, 3, 5, 7, 9). The optimum temperature of the recombinant enzymes was determined by monitoring the change in activity at different temperatures (25°C, 30°C, 35°C, 40°C, 45°C, 50°C, 60°C). The enzyme activity was measured as described above.

### 2.5.2. Effect of various metal ions on enzyme activity

Metal ions ($Fe^{2+}$, $Cu^{2+}$, $Mg^{2+}$, $Ca^{2+}$, $Zn^{2+}$) were added into the recombinant enzymes with different concentrations (0, 0.5, 1, 1.5, 2, 2.5 mM) and the enzyme activity was measured as described above. A control experiment without metal ions was also performed.

## 2.6. Degradation of different dyes with recombinant enzymes

The reaction system (3.5 ml) for decolourization contained dyes (final concentration 50 mg l$^{-1}$) and 500 μl recombinant enzymes (0.02 U). The azo dyes MY1 and CR, the anthraquinone dye disperse blue 2BLN and the triphenylmethane dye bromophenol blue were dissolved in the buffers (50 mM, pH = 5.0). The decolourization was carried out for 5 s at 30°C. The decolourization ability for each dye was evaluated by calculating the decrease in the maximum absorbance for each dye with the following equation:

$$\text{decolourization } (\%) = \frac{A_0 - A_t}{A_0} \times 100,$$

where $A_0$ is the initial absorbance of the dye and $A_t$ the absorbance of the dye with time.

The decolourization (%) reflected the decrease in the dye concentration because of the oxidation by recombinant enzymes.

## 2.7. Fourier-transform infrared and gas chromatograph-mass spectrometry analyses of metabolites

The assays were carried out at 30°C. The reaction mixture in a total volume of 10 ml contained dyes (MY1 or CR: final concentration of 50 mg l$^{-1}$), buffer (50 mM, pH = 5.0) and three recombinant enzymes (0.05 U). After degradation of the dye for 24 h using the enzyme, the dye degradation products were chromatographically extracted with pure ethyl acetate. Ethyl acetate was added to the sample and the mixture was condensed to 1 ml using a rotary evaporator. The condensed samples were then used for FTIR (Magna-IR 750, Thermo Nicolet) and GC-MS analyses. In addition, CR was degraded by a combined enzyme (Lac + MnP + LiP) and the metabolites were analysed by GC-MS as described above. The GC-MS analysis was performed using an Agilent 7890A gas chromatograph system coupled to an Agilent 5975C inert mass-selective detector with a Triple-Axis detector system (Agilent Technologies Inc., USA). Specific parameters were referenced to the results of Pan *et al.* [32].

## 2.8. Phytotoxicity assay

The phytotoxicity assays were performed to evaluate the toxicity of the CR dye, which has a complex structure, before and after degradation. Dye wastewater can contaminate irrigation water or be discharged into water bodies, affecting the growth of peppers. Thus, in this study, 50 pepper seeds were immersed in 75% alcohol for 5 min, washed with water, and then cultured. After the seeds were cultured for about a week, the growing pepper seedlings were transplanted into the CR dye (50 mg l$^{-1}$), the degraded CR dye and distilled water. The lengths of the leaf, shoot, root and the total weight after 15 days were recorded. All samples were incubated at the same environmental conditions. The experiment was repeated three times.

# 3. Results and discussion

## 3.1. Characterization and expression of recombinant enzymes' genes

The sequences were identified based on the genes in the NCBI database. The cloned MnP and LiP genes were found to be 73% and 87% similar to the MnP and LiP mRNA of *Phanerochaete chrysosporium*, respectively. The sequencing analysis revealed that the cloned sequence of Lac was similar to the Lac mRNA of *A. niger*, with 81% identity (electronic supplementary material, figure S1). After sequence assembly, the three enzyme genes were obtained and analysed by the open reading frame (ORF) Finder. The sequence analysis showed that the ORF of the Lac gene was 1827 bp and the gene encoded a protein of 608 amino acids. The ORF of the MnP gene was 1134 bp and the gene encoded a protein of 377 amino acids. The DNA sequence of the LiP gene was 1119 bp and the gene encoded a protein of 372 amino acids. The active centre of Lac from *A. niger* CBS was highly conserved and contained three cupredoxin domains at the T1 Cu binding site (H496, C580, H585) and the trinuclear Cu binding site (H125, H127, H169, H171). Similarly, three cupredoxin domains were also present in the active site of the recombinant Lac, including the T1 Cu binding site (H591, C592, H593) and the trinuclear Cu binding site (H125, H127, H180, H182). Diverse amino acid residues occurred near the active site, which might cause different catalytic performances of Lac from TS-A. The active centres of MnP and LiP from TS-A, such as the Mn binding site (E56, E60, D199) of MnP and the substrate binding site (R70, F73, H74) of LiP, were also highly conserved. A single colony was selected, and using PCR and agarose gel electrophoresis analyses, the transformant was verified to be successfully transformed (electronic supplementary material, figure S2). Ligninolitic enzyme sequences were uploaded using Dryad (https://doi.org/10.5061/dryad.4tmpg4f70) [33].

## 3.2. Determination of protein and enzyme activities

For the SDS-PAGE analysis of the recombinant enzymes, the gel was stained with Coomassie brilliant blue. The SDS-PAGE analysis results are shown in figure 1*d*. Most fungal Lacs are monomeric proteins, and LiPs and MnPs are monomeric haem-containing proteins [9,15]. Based on the literature, the typical fungal Lac is a 60–80 kDa molecule [34]. Moreover, the most common MnPs have a molecular weight of 32–62.5 kDa [7]. Chen *et al.* reported that the recombinant Il-MnP was monomeric

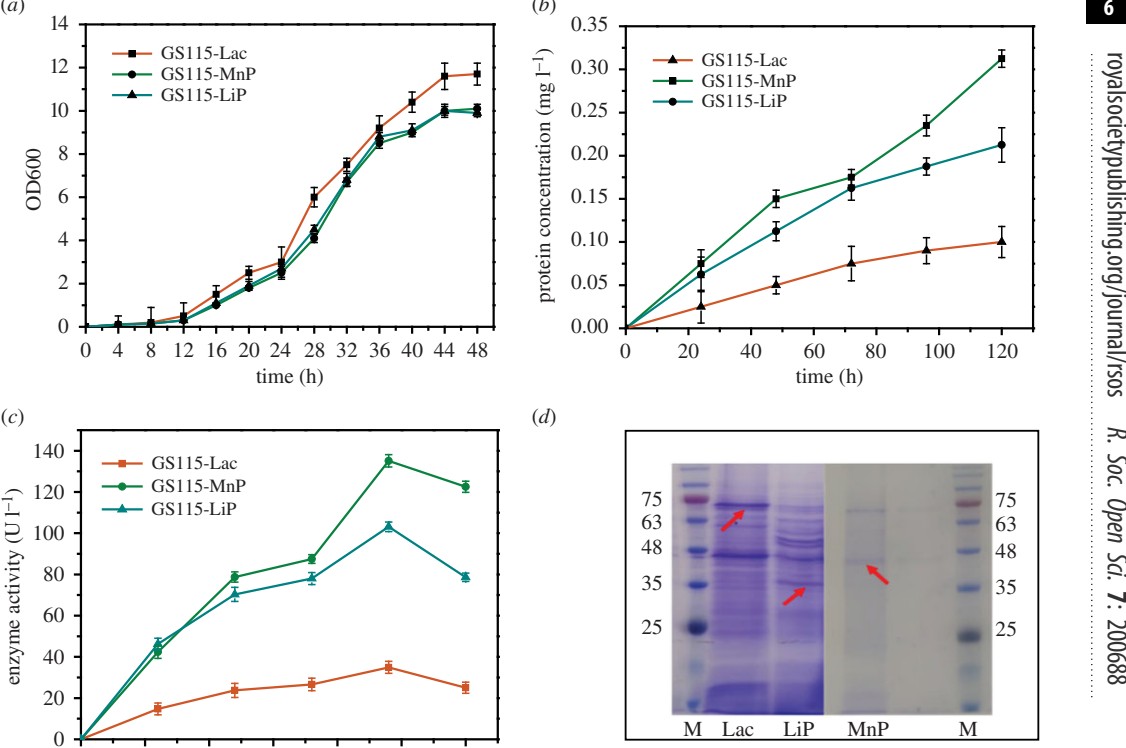

**Figure 1.** Determination of protein and enzyme activities: (*a*) OD600 of recombinant *Pichia pastoris* GS115; (*b*) changes in protein concentration in recombinant yeast fermentation; (*c*) activities of recombinant enzymes; and (*d*) SDS-PAGE results of the recombinant enzymes from *P. pastoris*, where M represents markers, and Lac, LiP and MnP represent the SDS-PAGE results of the recombinant enzymes from GS115-Lac, GS115-LiP and GS115-MnP, respectively.

and had a molecular weight of 43 kDa [35]. In addition, fungal LiPs are globular and mostly helical glycoproteins of about 40 kDa [9,25]; thus, the 69, 45 and 35 kDa bands obtained from the SDS-PAGE analysis correspond to the recombinant Lac, recombinant MnP and recombinant LiP, respectively.

The biomass production trend during a 48 h incubation period is illustrated in figure 1*a*. A rapid growth of the biomass was detected at the early stage of the GS115 fermentation process, and it reached the highest value at 44 h, suggesting that the transformant had advanced into the stationary growth stage. After culturing for 20–36 h in a logarithmic growth phase, the cells were collected and transferred to a BMMY medium to induce enzyme production. The protein concentration and enzyme activity of GS115-Lac, GS115-MnP and GS115-LiP reached a maximum of $0.1 \, mg \, ml^{-1}$ and $34 \, U \, l^{-1}$, $0.3 \, mg \, ml^{-1}$ and $135.12 \, U \, l^{-1}$ and $0.2 \, mg \, ml^{-1}$ and $103.13 \, U \, l^{-1}$, respectively (figure 1*b,c*). Compared with Lac and LiP, which had a low yield from the original strain, the recombinant enzymes and MnP had 50 and 10 times greater yields, respectively. The yields of the three enzymes were increased (electronic supplementary material, figure S3). With an increase in the fermentation time, in the late fermentation stage, the activities of the three enzymes were reduced, which may be caused by the influence of the accumulated product of the fermentation on the enzyme. In a previous study, similar results were obtained: the recombinant Lac produced by the expression of *Bacillus licheniformis* Lac cDNA in *P. pastoris* had the highest enzyme activity after seven days of fermentation, and the enzyme activity also decreased in the later stage [36].

The successfully expressed transformants were named GS115-Lac, GS115-MnP and GS115-LiP.

## 3.3. Characterization of recombinant enzymes

### 3.3.1. Effects of different temperatures and pH on recombinant enzymes

To find the temperature required for the optimum enzyme activity, the effects of different temperatures on the recombinant enzymes activities were determined (figure 2*a*). The recombinant MnP exhibited the

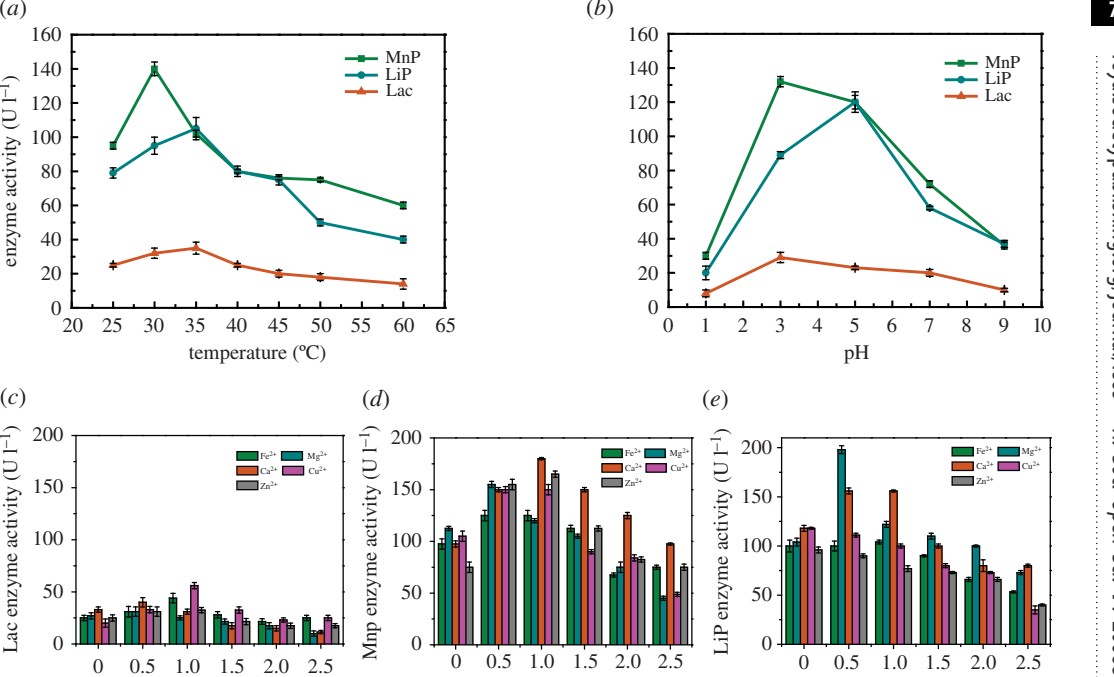

**Figure 2.** Effects of different temperatures, pH and metal ion concentrations on the activities of recombinant enzymes: (*a*) effects of different temperatures on recombinant enzymes; (*b*) effects of different pH on recombinant enzymes; (*c*) effects of different metal ion concentrations on the activities of Lac; (*d*) effects of different metal ion concentrations on the activities of MnP; and (*e*) effects of different metal ion concentrations on the activities of LiP.

highest enzyme activity (140 U l$^{-1}$) at 30°C; it showed a relatively strong activity at a broad temperature range of 25–50°C. Moreover, it retained 53.6% of the highest activity at 45°C and 42.9% at 60°C. Chen *et al.* reported that the recombinantly expressed MnP in *E. coli* showed 75% of the highest activity at 45°C and the enzyme activity decreased sharply at temperatures above 45°C [35]. Therefore, this recombinant MnP features strong ability at higher temperatures. The optimum temperature for the recombinant LiP was 35°C, and it showed a relatively strong activity at a broad temperature range of 25–45°C. The optimum temperature for the recombinant Lac activity was 35°C, corresponding to an enzyme activity of 35 U l$^{-1}$ and the recombinant Lac retained 42.9% of the highest activity at 60°C. In addition, the recombinant Lac showed a relatively strong activity at a broad temperature range of 25–60°C. The recombinant LiP, Lac and MnP showed optimum temperatures similar to those in the literature [17,35,36].

The optimum pH was observed for the three recombinant enzymes. The optimum pH values of the recombinant MnP, LiP and Lac were 3.0, 5.0 and 3.0, respectively (figure 2*b*). The three recombinant enzymes showed a relatively strong activity at a broad pH range of 3.0–7.0. This indicates that the enzymes can work under a wide pH range, especially in acidic conditions. The pH range of the three recombinant enzymes was similar to that reported for ligninolytic enzymes secreted by a fungus, and most of the optimum pH values were 4.0 and 6.0 [7,37].

### 3.3.2. Effects of different metal ions on recombinant enzymes

After incubation with different metal ions, $Cu^{2+}$ exhibited the most stimulatory effect on the recombinant Lac activity (figure 2*c*). The activity of the enzyme with 1 mM $Cu^{2+}$ was 56 U l$^{-1}$, higher than that of the control (20 U l$^{-1}$). It has been reported that the addition of $Cu^{2+}$ to the purified enzyme stimulated Lac activity, but mostly at low concentrations of $Cu^{2+}$, lower than 1 mM [38]. As Lac is a copper-containing polyphenol oxidase, its activity stimulation may be regulated by copper ions [29]. Moreover, $Fe^{2+}$ also stimulated the recombinant Lac activity, with activities of 45 U l for 1 mM $Fe^{2+}$, and $Mg^{2+}$, $Ca^{2+}$ and $Zn^{2+}$ at a concentration of 0.5 mM could promote the Lac activity. Conversely, 2.5 mM $Mg^{2+}$ slightly inhibited Lac activity, and $Ca^{2+}$ had a significant effect on the MnP enzyme activity (figure 2*d*), which corresponds to the reported results [35]. The LiP activities (figure 2*e*) in the presence of 0.5 mM $Mg^{2+}$

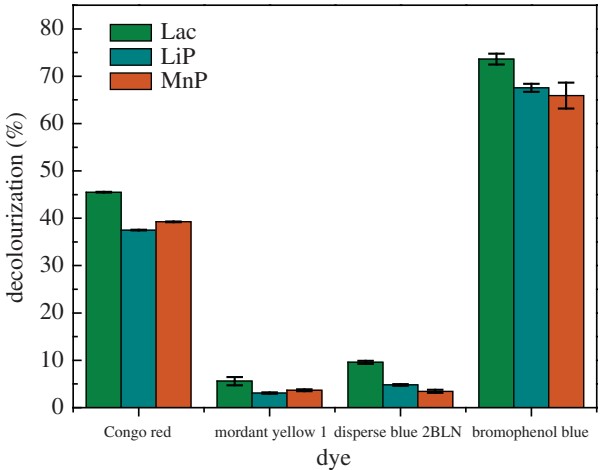

**Figure 3.** Recombinant enzymes degradation of different dyes.

and $Ca^{2+}$ were 198 and 158 U $l^{-1}$, higher than that of the control (120 U $l^{-1}$), and the other three metal ions inhibited the LiP activity.

## 3.4. Degradation of different dyes with recombinant enzymes

The results are illustrated in figure 3. Lac, LiP and MnP exhibited a certain decolourization ability for dyes with different structures. For the three enzymes, the decolourization rates of CR were higher than those of MY1, by up to 45.5% in 5 s. The enzymes had no significant effect on the decolourization of the anthraquinone dye disperse blue 2BLN. The three enzymes had the most significant effect on the triphenylmethane dye bromophenol blue; the decolourization rate could reach 73.6% in 5 s. In this work, the decolourization rate of the acid dye bromophenol blue [39] was the highest among these dyes. This corresponds to the optimal pH of the recombinant enzyme in §3.3.1. Other researchers have also reported the degradation of acid dyes. Sun *et al.* reported that methyl orange was decolourized by Lac obtained from *G. lucidum* by about 57.48% in the presence of the redox mediator ABTS [40]. Moreover, purified recombinant Lac has been reported to decolourize 98.1% of CR and 98.5% of malachite green in about 3 h in the presence of the natural redox mediator acetosyringone [41]. In addition, in the current study, the recombinant enzyme showed a fast decolourization ability for different dyes.

## 3.5. Degradation pathway of azo dyes by recombinant enzymes

The recombinant enzymes were efficient in the decolourization of MY1 and CR. After 24 h, the decolourization rates of MY1 by the recombinant Lac, LiP and MnP were 67%, 58% and 54%, respectively, and those of CR were 89%, 91% and 90%, respectively (electronic supplementary material, figure S4). To understand the transformations of the azo dyes catalysed by the recombinant enzymes, two analytical tools were used: FTIR and GC-MS.

### 3.5.1. Fourier-transform infrared analysis

The recombinant enzymes were used to degrade the MY1 and CR azo dyes, and the degradation products were analysed using FTIR. The characteristic absorption peaks of the MY1 and CR molecular structures are shown in figure 4. The first group of MY1 peaks featured a peak at 3440 $cm^{-1}$, attributed to the N–H stretching of amine [42]. The second group of MY1 peaks included a peak at 1617 $cm^{-1}$, corresponding to N=N stretching; 1532 $cm^{-1}$, for aromatic C=C stretching; 1357 $cm^{-1}$, for C–N stretching and 1078 $cm^{-1}$, for C–O stretching. The third group of peaks ranged from 900 to 600 $cm^{-1}$, with peaks at 810 and 739 $cm^{-1}$, for N–H stretching, and at 699 $cm^{-1}$, for the C–H stretching in benzene. The FTIR profile of the untreated CR dye showed various functional groups: the 3448 $cm^{-1}$ peak represents the N–H stretching of amine; 1637 $cm^{-1}$ represents N=N stretching, 1541 $cm^{-1}$ represents aromatic C=C stretching and 1117 $cm^{-1}$ represents –C–S–. The characteristic peaks indicate that the two dyes were azo dyes.

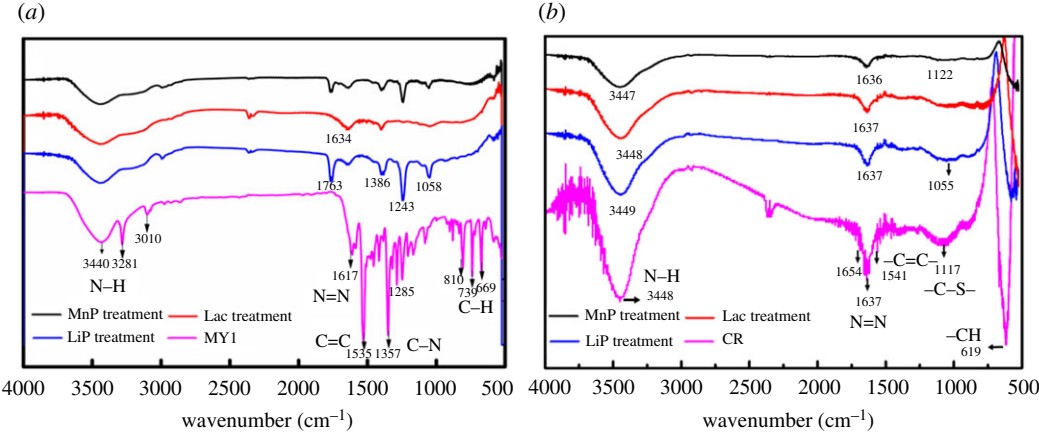

**Figure 4.** FTIR spectra of azo dyes and their degradation products during the degradation by recombinant enzymes: (*a*) MY1 and degradation products by LiP, Lac and MnP; and (*b*) CR and degradation products by LiP, Lac and MnP.

In the biodegradation of MY1 with LiP, the disappearance of the peak at $1617 \text{ cm}^{-1}$ showed that the azo group was cleaved. The peaks at 1535, 3010 and $3281 \text{ cm}^{-1}$ disappeared, which indicates that LiP was involved in the N=N cleavage, and the disappearance of the peak at $669 \text{ cm}^{-1}$ indicates that LiP participated in the ring-opening reaction. Moreover, the analysis results of the degradation products showed minor peaks at 1058 and $1243 \text{ cm}^{-1}$ (figure 4*a*), which represent C–H stretching and C–O stretching, respectively; these peaks indicate that the LiP could degrade MY1. For CR and the degradation products, the stretching peak at $1637 \text{ cm}^{-1}$ weakened, and minor peaks appeared at $1055 \text{ cm}^{-1}$ (figure 4*b*), corresponding to C–H stretching, indicating that the LiP could degrade CR. Figure 4*a* demonstrates that Lac could destroy N=N. The peak at $669 \text{ cm}^{-1}$ disappeared, indicating that Lac could also open the benzene ring. Moreover, the peak of benzene-containing compounds did not appear in the results. This shows that the MY1 degradation by Lac was more complete and the Lac recombinant enzyme also exhibited the highest degradation. Figure 4*b* proves that the stretching peak at $1637 \text{ cm}^{-1}$ weakened and Lac could destroy N=N in CR. The spectra of MY1 degradation by MnP is shown in figure 4*a*. The appearance of the peak is similar to the degradation by Lac treatment and MnP could destroy N=N. MnP could also destroy N=N in CR, as shown in figure 4*b*, and the stretching peak at $1636 \text{ cm}^{-1}$ weakened. The FTIR spectrum of the treated dye after biodegradation with the recombinant indicates that the azo dye degradation was initiated by the destruction of the N=N bond and then mineralization. The recombinant enzymes could degrade azo dyes, whether monoazo or diazo dyes.

### 3.5.2. Gas chromatography-mass spectrometry analysis

In this study, GC-MS was also used to analyse the degradation products and confirm the pathways of MY1 and CR removal by the three enzymes. The dye degradation products of the three enzymes had low molecular weights. It can be seen from the infrared spectra in figure 4 that the azo bonds in MY1 and CR dyes were significantly degraded.

Table 2 presents the degradation products during the MY1 degradation process by the recombinant enzymes. The GC-MS result showed the mass spectra of nine different degradation products. The compound methyl 2-fluoro-5-nitrobenzoate was identified to have the highest molecular weight, while benzene, 2-methylpropionic acid and 3-methyl-1-butanol had the smallest weights. The three enzymes all yielded *p*-xylene, *o*-xylene, *m*-xylene and benzene as part of the intermediates from the MY1 degradation. Each enzyme also had unique degradation products: phenylethyl alcohol and 3-methyl-1-butanol were detected during the degradation with MnP; butanoic acid and 3-methyl- were detected during the degradation with LiP and 2-methylpropionic acid was detected during the degradation with Lac. The results indicate that the products of degradation using LiP and Lac were usually acidic [43,44]. It was proved that all three enzymes could degrade MY1, and the degradation products were mostly of low molecular weight and low toxicity.

Table 3 presents the CR degradation products obtained using the recombinant enzymes. Because of the complex molecular structure of CR, GC-MS showed the mass spectra of 25 different degradation products. The degradation intermediates contained amine groups, including pyrrolo

**Table 2.** Biodegradation products of mordant yellow 1 by MnP, LiP and Lac, as obtained through GC-MS analysis (50 mg l$^{-1}$ mordant yellow 1).

| number | metabolite | molecular weight | retention time (min) | chemical formula | catalytic system |
|---|---|---|---|---|---|
| 1 | methyl 2-fluoro-5-nitrobenzoate | 185.11 | 27.88 | $C_7H_4FNO_4$ | LiP, Lac |
| 2 | phenylethyl alcohol | 122.17 | 18.28 | $C_8H_{10}O$ | MnP |
| 3 | *p*-xylene | 106.167 | 7.45 | $C_8H_{10}$ | LiP, MnP, Lac |
| 4 | *o*-xylene | 106.16 | 6.42 | $C_6H_4(CH_3)_2$ | LiP, MnP, Lac |
| 5 | *m*-xylene | 106 | 6.11 | $C_8H_{10}$ | LiP, MnP, Lac |
| 6 | butanoic acid, 3-methyl- | 102.13 | 5.97 | $C_5H_{10}O_2$ | LiP |
| 7 | benzene | 78.11 | 3.51 | $C_6H_6$ | LiP, MnP, Lac |
| 8 | 2-methylpropionic acid | 88.11 | 3.37 | $C_4H_8O_2$ | Lac |
| 9 | 3-methyl-1-butanol | 88.15 | 2.97 | $C_5H_{12}O$ | MnP |

[1, 2-a] pyrazine-1, 4-dione, hexahydro-3-(phenylmethyl)- and tryptophol. Similar metabolites from CR degradation using ligninolytic enzymes from *A. niger* have been obtained in a previous study [12]. In the current study, the degradation intermediates also included benzeneacetic acid, phenylethyl alcohol, *p*-xylene and *o*-xylene, which had fewer benzene-ring structures and reduced toxicity. The products with low molecular weight and open benzene rings detected from CR degradation included 3-methyl-butanoic acid and 2-methyl-butanoic acid, and their toxicity was significantly decreased. Similarly, Iark *et al*. also reported that open benzene-ring compounds and a fully oxygenated compound were obtained from Lac degraded CR and that the toxicity was reduced [19]. These degradation products prove that the three enzymes can degrade CR and reduce toxicity by breaking azo bonds and opening the benzene ring. To simulate the complex enzyme system of the original strain *Aspergillus* sp. TS-A, the three enzymes were combined to degrade CR. Surprisingly, some new products such as 3, 5-dimethyl-4-heptanone and 3-(methylthio)-propanoic acid, which did not exist in the degradation products obtained using the individual enzymes appeared in the products obtained by the combined enzyme. Combining the three enzymes might result in a more complete degradation of CR.

The three enzymes degraded not only monoazo dyes but also diazo dyes. Moreover, some of the MY1 and CR degradation products obtained by separately using the three enzymes were the same, which suggests that the degradation pathways of the two dyes by the three enzymes may be similar.

### 3.5.3. Degradation pathways of azo dyes

Based on the degradation products detected by GC-MS, the possible degradation pathways of the three recombinant enzymes on the azo dye were analysed. For the MY1 degradation, the possible degradation path is shown in figure 5a. From the degradation products of Lac, first, the Lac-catalysed azo bonds (–N=N–) were cleaved, and then the hydroxy group of the ortho carboxyl group was replaced by F–; then the benzene ring of compounds finally broke, and the compounds degraded into less-toxic metabolites, such as 2-methyl-propionic acid. From the degradation products of LiP (figure 5), first, the azo bonds between the benzene rings were cleaved, followed by a carboxylation reaction to substitute the benzene branch, and finally the degradation into less toxic and small molecules metabolites, such as 3-methyl-butanoic acid. Regarding the degradation products of MnP, the azo bond was opened, but nitrogen-containing compounds were not detected as intermediate products; the last reaction was similar to those of the degradations using Lac and LiP, and less-toxic products compared with the original dye were detected, such as 3-methyl-1-butanol. Furthermore, MnP has been reported to effectively cleave benzene rings of the phenol-containing organic compound nonylphenol, and the final products were mostly alcohols [45]. Yang *et al*. [46] reported that MnP produced by white-rot fungus *Irpex lacteus* F17 could cause the oxidative cleavage of the C = C bond. That is, the three enzymes showed strong degradation and mineralization ability of MY1, which provides an effective way to treat effluents containing monoazo dye.

**Table 3.** Biodegradation product of Congo red by MnP, LiP and Lac, as obtained through GC-MS analysis (50 mg l$^{-1}$ Congo red) (combined: LiP + MnP + Lac); not all metabolites are shown.

| number | metabolite | molecular weight | retention time (min) | chemical formula | catalytic system |
|---|---|---|---|---|---|
| 1 | l-proline, N-allyloxycarbonyl-, tetradecyl ester | 395.303 | 38.892 | $C_{23}H_{41}NO_4$ | LiP, combined |
| 2 | bis(2-ethylhexyl) phthalate | 390.303 | 35.599 | $C_{24}H_{38}O_4$ | combined |
| 3 | 9-octadecenamide, (Z)- | 281.272 | 37.391 | $C_{18}H_{35}NO$ | LiP, Lac |
| 4 | benzeneacetic acid, decyl ester | 276.208 | 12.958 | $C_{18}H_{28}O_2$ | Lac |
| 5 | pyrrolo[1,2-a]pyrazine-1,4-dione, hexahydro-3-(phenylmethyl)- | 244.121 | 34.014 | $C_{14}H_{16}N_2O_2$ | LiP, MnP, Lac, combined |
| 6 | diphenylolpropane | 228.115 | 31.834 | $C_{15}H_{16}O_2$ | Lac |
| 7 | pyrrolo[1,2-a]pyrazine-1,4-dione, hexahydro-3-(2-methylpropyl)- | 210.136 | 27.429 | $C_{11}H_{18}N_2O_2$ | MnP, Lac |
| 8 | tryptophol | 161.084 | 24.324 | $C_{10}H_{11}NO$ | MnP, combined |
| 9 | diethyltrisulfide | 153.994 | 25.508 | $C_4H_{10}S_3$ | MnP, Lac |
| 10 | propanoic acid, 2-methyl-, butyl ester | 144.115 | 5.164 | $C_8H_{16}O_2$ | LiP |
| 11 | butanoic acid, 2-methylpropyl ester | 144.115 | 5.209 | $C_8H_{16}O_2$ | LiP, MnP |
| 12 | butanoic acid, butyl ester | 144.115 | 5.985 | $C_8H_{16}O_2$ | LiP, MnP, Lac |
| 13 | hexanoic acid, 2-oxo-, methyl ester | 144.078 | 4.905 | $C_7H_{12}O_3$ | MnP |
| 14 | 2,2-dimethyl-3-heptanone | 142.135 | 4.808 | $C_9H_{18}O$ | MnP |
| 15 | 3,5-dimethyl-4-heptanone | 142.135 | 4.782 | $C_9H_{18}O$ | combined |
| 16 | benzeneacetic acid | 136.052 | 12.473 | $C_8H_8O_2$ | LiP, MnP, Lac, combined |
| 17 | 4-methyl-2-oxovaleric acid | 130.063 | 4.84 | $C_6H_{10}O_3$ | LiP, combined |
| 18 | nonane | 128.157 | 3.436 | $C_9H_{20}$ | MnP, combined |
| 19 | phenylethyl alcohol | 122.073 | 8.767 | $C_8H_{10}O$ | LiP, MnP, Lac, combined |
| 20 | propanoic acid, 3-(methylthio)- | 120.025 | 8.592 | $C_4H_8O_2S$ | combined |
| 21 | p-xylene | 106.678 | 3.889 | $C_8H_{10}$ | LiP, MnP, Lac, combined |
| 22 | o-xylene | 106.678 | 4.213 | $C_8H_{10}$ | LiP, MnP, Lac, combined |
| 23 | ethylbenzene | 106.078 | 3.792 | $C_8H_{10}$ | LiP, MnP, Lac, combined |
| 24 | butanoic acid, 3-methyl- | 102.068 | 3.838 | $C_5H_{10}O_2$ | LiP, MnP, combined |
| 25 | butanoic acid, 2-methyl- | 102.068 | 4.458 | $C_5H_{10}O_2$ | MnP, Lac, combined |

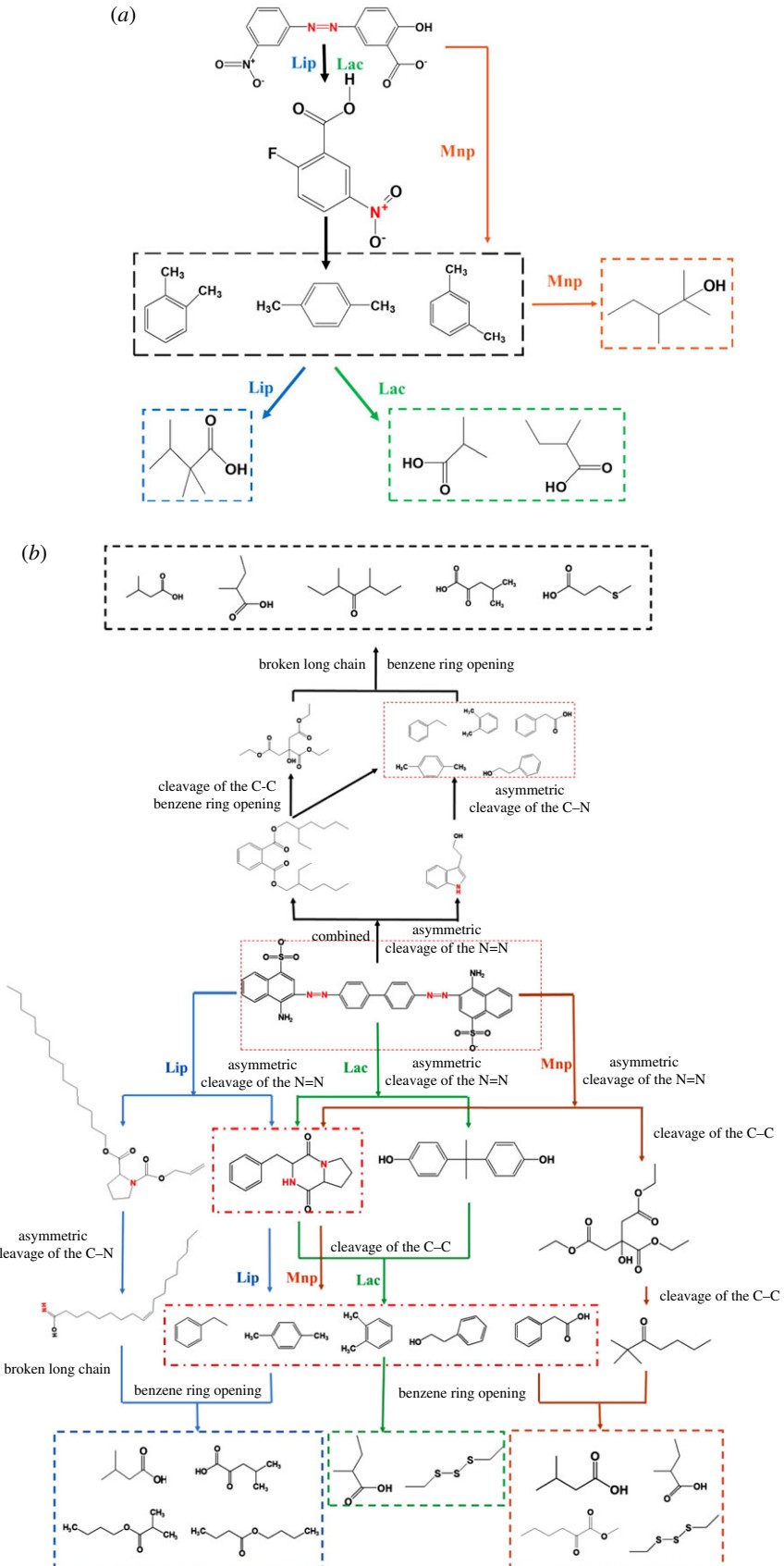

**Figure 5.** Possible degradation pathways of azo dyes degraded by MnP, LiP and Lac, as obtained through GC-MS analysis. (*a*) Possible degradation pathways of MY1 by MnP, LiP and Lac; and (*b*) possible degradation pathways of Congo red by MnP, LiP, Lac and combined; combined: LiP + MnP + Lac.

**Table 4.** Phytotoxicity of CR on pepper seedlings before and after biodegradation by Lac, LiP and MnP. (The different letters indicate significant differences among the conditions, $p < 0.05$ (combined: LiP + MnP + Lac).)

| parameters | water | CR | CR degraded by Lac | CR degraded by LiP | CR degraded by MnP | CR degraded by combined |
|---|---|---|---|---|---|---|
| leaf length (cm) | 2.40 ± 0.08a | 1.33 ± 0.12c | 1.97 ± 0.12b | 2.33 ± 0.12a | 1.83 ± 0.17b | 2.36 ± 0.09a |
| shoot length (cm) | 7.97 ± 0.37a | 5.10 ± 0.14b | 6.40 ± 0.43b | 8.37 ± 1.39a | 5.13 ± 0.19b | 8.00 ± 1.41a |
| root length (cm) | 6.50 ± 0.57a | 3.67 ± 0.24c | 5.00 ± 0.81b | 6.57 ± 0.49a | 5.56 ± 0.09ab | 6.46 ± 0.61a |
| weight (g) | 0.19 ± 0.01a | 0.14 ± 0.01a | 0.17 ± 0.01a | 0.21 ± 0.04a | 0.19 ± 0.01a | 0.20 ± 0.01a |

For CR degradation, the possible degradation pathway is shown in figure 5*b*. The CR degradation by LiP may occur via the following steps: (i) the asymmetric cleavage of the N=N bonds, pyrrolo [1,2-a] pyrazine-1, 4-dione, hexahydro-3-(phenylmethyl), and l-proline-N-allyloxycarbonyl-tetradecyl ester were detected; (ii) the asymmetric cleavage of the C–N bond, removing N in the aromatic ring, and yielding several intermediate products, such as 9-octadecenamide, (Z)- and phenylethyl alcohol, and *p*-xylene; and (iii) the breakage of benzene ring opening and the transformation of long-chain compounds into low-molecular-weight and less-toxic stable products, such as butanoic acid, 3-methyl- and 4-methyl-2-oxovaleric acid. The azo and $SO_3$ groups were lost in degradation. Some of these results were found in previous studies. For example, in one study, LiP could mineralize various recalcitrant aromatic and halogenated phenolic compounds [24]. A similar CR degradation pathway has also been reported [12]. The degradation pathway using MnP was different, in that dibutyl phthalate was detected after the asymmetric cleavage of the N=N bond. Here, after the benzene ring was broken, dibutyl phthalate was converted to 1, 9-dioxacyclohexadeca-4,13-diene-2-10-dione,7,8,15,16-tetramethyl and then broken to transform into triethyl citrate. Finally, the intermediates turned into low-molecular-weight and less-toxic stable degraded products. During the degradation of CR by MnP, some new less-toxic molecules were detected, such as tryptophol, diethyltrisulfide and 2-methyl-butanoic acid. Diethyltrisulfide could be obtained from the $SO_3$ group in the CR. The degradation pathway of Lac was similar to that of MnP, and diphenylolpropane was detected, which proves that the CR dye was degraded and mineralized by Lac. The degradation pathway may be as follows: first, the asymmetric cleavage of one N=N bond occurs, then the other N=N bonds are broken. Previous studies have reported that Lac could catalyse the ring cleavage reaction of aromatic compounds [14,15]. In this work, small molecules such as 2-methyl-butanoic acid and diethyltrisulfide were detected, which had open benzene rings, and the broken long chains were converted into stable and less-toxic products. For CR degradation using Lac, Iark *et al.* reported a similar pathway involving asymmetric cleavage [19]. This current work proves that the three recombinant enzymes could degrade diazo dyes. When they were combined to degrade CR, the key pathways were similar to those of the degradation using the individual enzymes. However, some new less-toxic small molecules appeared during the degradation with the combined enzymes, such as 3-(methylthio)-propanoic acid and 3, 5-dimethyl-4-heptanone. Moreover, the combined ligninolytic enzymes exhibited more degradation products of diazo dyes compared with those of the single enzymes. This suggests that the co-degradation of the combined enzyme made the dye degradation more complete. Therefore, the individual three enzymes and a combination of them could degrade and mineralize CR dyes. The combined enzyme system has a potential for application in dye wastewater treatment.

## 3.6. Phytotoxicity assay

The decolourization rate of CR degraded by the recombinant enzymes could reach 90% after 24 h (electronic supplementary material, figure S4). As shown in table 4, the CR dye had significantly lower biological properties of pepper seedlings compared with those of the water-treated seedlings. Previous studies have reported that phytotoxicity trials on *Vigna radiata* confirmed the toxic nature of the untreated dye solutions [47]. Moreover, the parameters of pepper seedlings treated with CR degraded by Lac, LiP and MnP were close to those of the water-treated seedlings and significantly higher than those of the CR-treated seedlings. Therefore, the degradation products of CR were substantially less toxic and probably non-toxic. The recombinant enzymes may also have a

detoxification ability on the MY1 dye, as its degradation pathways and products were similar to those of CR.

# 4. Conclusion

In summary, Lac, MnP and LiP genes from *Aspergillus* sp. TS-A were cloned and expressed in *P. pastoris*, a system that afforded a relatively high enzyme production. The three recombinant enzymes were found to exhibit strong activities over a wide range of pH and temperature, and the recombinant enzymes showed enhanced activity in the presence of certain metal ions. In addition, the recombinant enzymes also had a fast decolourization ability. The analysis of metabolites proved the degradation and detoxification of the azo dye, and the possible degradation pathways were estimated. The combination of the three recombinant enzymes could provide a highly efficient catalytic reaction system.

Data accessibility. The data supporting this article have been uploaded as part of the electronic supplementary material, word file. Data has also been deposited in Dryad: https://doi.org/10.5061/dryad.4tmpg4f70 [33].

Authors' contributions. S.L. carried out experiments and wrote the manuscript; X.X. guided the experiment and writing of the manuscript; Y.K., Y.X. and H.L. assisted in the experiments. All authors gave final approval for publication.

Competing interests. The authors declare no competing interests.

Funding. This work was supported by the National Natural Science Foundation of China (grant no. 21466032) and Scientific Funding Research Foundation for Chang Jiang Scholars of Shihezi University (grant no. CJXZ201501).

Acknowledgements. We thank Qikai Fu, Deliang Guo, Zhaonan Yang and Yulong Zhou for assistance with this study.

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
