## [Reviewer comments · Royal Society Open Science]

Review History

RSOS-200688.R0 (Original submission)

Review form: Reviewer 1

Is the manuscript scientifically sound in its present form?

Yes

Are the interpretations and conclusions justified by the results?

Yes

Is the language acceptable?

Yes

Do you have any ethical concerns with this paper?

No

Have you any concerns about statistical analyses in this paper?

No

Recommendation?

Major revision is needed (please make suggestions in comments)

Comments to the Author(s)

The Ms deals with the degradation of synthetic dyes by cloning ligninolytic enzymes, i.e. Lac, Mnp and lip, from *Aspergillus* sp. to *Pichia pastoris* in 3 plasmids.

Degradation of dyes by ligninolytic enzymes is not new, and the authors mentioned some papers related to this topic. However, the Ms showed interesting results.

But I have some queries to be answered:

- Are the sequences of the 3 enzymes publically available in some website? Since the authors isolated the genes of the 3 ligninolytic enzymes from an *Aspergillus* spp.

- The aim of the work was to degrade dyes partially or totally, means potential use at large scale. So, how stable are the plasmids in *P. pastoris*? (How many generations? How many batches? Did the authors try to scale up the process?)

- In a complex mixture of chemical pollutants (e.g. more than one toxic molecule commonly available in tannery wastewaters, how the authors are able to combine the activities of 3 enzymes?)

- Why the toxicity assays was performed in pepper seeds? Since Congo red is considered carcinogenic, and the result showed (decolorization, %) were from 45 to 38%? Some common mammalian cell line should be tested (e.g. normal human fibroblasts).

Review form: Reviewer 2

Is the manuscript scientifically sound in its present form?

Yes

Are the interpretations and conclusions justified by the results?

Yes

Is the language acceptable?

No

Do you have any ethical concerns with this paper?

No

Have you any concerns about statistical analyses in this paper?

No

Recommendation?

Accept with minor revision (please list in comments)

Comments to the Author(s)

A linguistic and grammatical revision of all work is strongly recommended.

Abstract:

- The term "specific activity" is misused because activity per mg protein was not reported.

- The term "decolorization ratio" was used instead of "rate".

Materials and Methods

- The meaning of the acronym for microbiological culture media should be written.

- According to the description of the study of pH effect on enzyme stability (2.5.1), stability is measured for 30 min and then the substrate was added. If the pH is not changed to the reference value of activity assay, activity is being measured at different pHs in each case. In this way, two studies overlap (stability/optimum pH) and should be carried out separately. Some details of the procedure are probably missing to help to understand it.

- Incubation time at each temperature is not reported in the thermostability study (2.5.1). It can be assumed that it is the time of the activity assay so two effects are being measured simultaneously and it is not a stability study per se.
- It would be important to report how much activity is added to the reaction system (2.6) in 0.5mL of enzyme solution in each case, or how many mg of protein and the specific activity of the sample.
- In 2.6 the term "concentration" is used to express Decolorization but it is not explained how it is calculated. Was absorbance correlated with a standard?
- In 2.7, what reaction volume was used for the analysis of the degradation products?
- Why was the pepper seed bioassay selected for the toxicity study? A reference is needed for this study (2.9).

Results

- It cannot be deduced molecular mass of a protein from an SDS-PAGE unless it can be said it is monomeric based on literature or another complementary study (See in 3.2 first paragraph). How were the bands corresponding to the enzymes of interest selected from among all the bands seen in the electrophoresis in fig.1d? One could refer to the literature, for example by citing reference 33 to say that the selected band corresponds to the laccase, but not make the deduction backwards, as was discussed. In addition, in that review several native enzymes are shown, which one is referenced in particular?
 - In fig 1 it would be suitable to clarify what the Lac, Mnp and Lip labels correspond to. It seems appropriate to name them as described in the text: GS115-Lac, GS115-Mnp, GS115-Lip.
 - In 3.2 line 35-36 the term recombinase is used wrongly.
 - In 3.3.1 line 48, references 34 and 35 are cited but in these works no thermostability of the LiP is reported.
 - Lines 49-50, stability and optimum pH concepts are confused.
 - Line 52, pH 2.0 cannot be included in the observation because this point was not tested for the recombinant strain.
- Furthermore, it is not valid to compare the results obtained for native and recombinant enzymes since solutions with different concentrations of active enzyme were used. If possible, equal protein concentration and equal enzymatic activity per unit volume should be used. Ref 36 cannot be used to discuss 3 lignin oxidases because this is a study of MnP only.

- I wonder if the resulting pH in the reaction medium is known or was measured (3.4).
- In 3.4, line 42-43, MO dye decolorization reported by Jing Sun et al. was done by laccase from *G. lucidum* in presence of the redox mediator ABTS 0,025mM, not directly as mentioned.
- It would be good to know the value of the decolorization at 24h of MY and CR to correlate it with the results of FTIR and GCMS.
- The retention times of the compounds detected in the GC-MS could be reported.
- Legend in figure 5 must be corrected.
- In 3.5.3 figure 6 is mentioned but there is no figure 6 in the document.
- It should be clarified what the abbreviation "com" means in Table 4.

Supplementary_material

- The legend in figure S1 must be corrected, see the order of the three enzymes results.
- In figure S4 the legend should be corrected too, because the result shown is not from the recombinant enzymes.

Decision letter (RSOS-200688.R0)

Dear Dr Xu:

Title: Degradation and Detoxification of azo dyes with recombinant ligninolytic enzymes from *Aspergillus* sp. with secretory overexpression in *P.pastoris*
Manuscript ID: RSOS-200688

The editor assigned to your manuscript has now received comments from reviewers. We would like you to revise your paper in accordance with the referee and Subject Editor suggestions which can be found below (not including confidential reports to the Editor). Please note this decision does not guarantee eventual acceptance.

Please submit your revised paper before 05-Jul-2020. Please note that the revision deadline will expire at 00.00am on this date. If we do not hear from you within this time then it will be assumed that the paper has been withdrawn. In exceptional circumstances, extensions may be possible if agreed with the Editorial Office in advance. We do not allow multiple rounds of revision so we urge you to make every effort to fully address all of the comments at this stage. If deemed necessary by the Editors, your manuscript will be sent back to one or more of the original reviewers for assessment. If the original reviewers are not available we may invite new reviewers.

On behalf of the Subject Editor Professor Anthony Stace and the Associate Editor Dr Nadia Martinez Villegas.

RSC Associate Editor:
Comments to the Author:

This research contributes to a deeper understanding of the degradation mechanism of three ligninolytic enzymes. While the conclusions are supported by the data, a serious problem with the English needs to be solved. In many cases it is difficult to understand what is meant and can lead to misunderstandings that affect the validity of the presented results. Finally, it is highly recommended to place the sequences of the ligninolytic enzymes and well as the *Aspergillus* spp. in some repository.

RSC Subject Editor:

Comments to the Author:
(There are no comments.)

Reviewers' Comments to Author:

Reviewer: 1

Comments to the Author(s)

The Ms deals with the degradation of synthetic dyes by cloning ligninolytic enzymes, i.e. Lac, Mnp and lip, from *Aspergillus* sp. to *Pichia pastoris* in 3 plasmids.

Degradation of dyes by ligninolytic enzymes is not new, and the authors mentioned some papers related to this topic. However, the Ms showed interesting results.

But I have some queries to be answered:

- Are the sequences of the 3 enzymes publically available in some website? Since the authors isolated the genes of the 3 ligninolytic enzymes from an *Aspergillus* spp.
- The aim of the work was to degrade dyes partially or totally, means potential use at large scale. So, how stable are the plasmids in *P. pastoris*? (How many generations? How many batches? Did the authors try to scale up the process?)
- In a complex mixture of chemical pollutants (e.g. more than one toxic molecule commonly available in tannery wastewaters, how the authors are able to combine the activities of 3 enzymes?
- Why the toxicity assays was performed in pepper seeds? Since Congo red is considered carcinogenic, and the result showed (decolorization, %) were from 45 to 38%? Some common mammalian cell line should be tested (e.g. normal human fibroblasts).

Reviewer: 2

Comments to the Author(s)

A linguistic and grammatical revision of all work is strongly recommended.

Abstract:

- The term "specific activity" is misused because activity per mg protein was not reported.
- The term "decolorization ratio" was used instead of "rate".

Materials and Methods

- The meaning of the acronym for microbiological culture media should be written.
- According to the description of the study of pH effect on enzyme stability (2.5.1), stability is measured for 30 min and then the substrate was added. If the pH is not changed to the reference value of activity assay, activity is being measured at different pHs in each case. In this way, two studies overlap (stability/optimum pH) and should be carried out separately. Some details of the procedure are probably missing to help to understand it.
- Incubation time at each temperature is not reported in the thermostability study (2.5.1). It can be assumed that it is the time of the activity assay so two effects are being measured simultaneously and it is not a stability study per se.
- It would be important to report how much activity is added to the reaction system (2.6) in 0.5mL of enzyme solution in each case, or how many mg of protein and the specific activity of the sample.

- In 2.6 the term "concentration" is used to express Decolorization but it is not explained how it is calculated. Was absorbance correlated with a standard?
- In 2.7, what reaction volume was used for the analysis of the degradation products?
- Why was the pepper seed bioassay selected for the toxicity study? A reference is needed for this study (2.9).

Results

- It cannot be deduced molecular mass of a protein from an SDS-PAGE unless it can be said it is monomeric based on literature or another complementary study (See in 3.2 first paragraph). How were the bands corresponding to the enzymes of interest selected from among all the bands seen in the electrophoresis in fig.1d? One could refer to the literature, for example by citing reference 33 to say that the selected band corresponds to the laccase, but not make the deduction backwards, as was discussed. In addition, in that review several native enzymes are shown, which one is referenced in particular?
- In fig 1 it would be suitable to clarify what the Lac, Mnp and Lip labels correspond to. It seems appropriate to name them as described in the text: GS115-Lac, GS115-Mnp, GS115-Lip.
- In 3.2 line 35-36 the term recombinase is used wrongly.
- In 3.3.1 line 48, references 34 and 35 are cited but in these works no thermostability of the LiP is reported.
- Lines 49-50, stability and optimum pH concepts are confused.
- Line 52, pH 2.0 cannot be included in the observation because this point was not tested for the recombinant strain.

Furthermore, it is not valid to compare the results obtained for native and recombinant enzymes since solutions with different concentrations of active enzyme were used. If possible, equal protein concentration and equal enzymatic activity per unit volume should be used.

Ref 36 cannot be used to discuss 3 lignin oxidases because this is a study of MnP only.

- I wonder if the resulting pH in the reaction medium is known or was measured (3.4).
- In 3.4, line 42-43, MO dye decolorization reported by Jing Sun et al. was done by laccase from *G. lucidum* in presence of the redox mediator ABTS 0,025mM, not directly as mentioned.
- It would be good to know the value of the decolorization at 24h of MY and CR to correlate it with the results of FTIR and GCMS.
- The retention times of the compounds detected in the GC-MS could be reported.
- Legend in figure 5 must be corrected.
- In 3.5.3 figure 6 is mentioned but there is no figure 6 in the document.
- It should be clarified what the abbreviation "com" means in Table 4.

Supplementary_material

- The legend in figure S1 must be corrected, see the order of the three enzymes results.
- In figure S4 the legend should be corrected too, because the result shown is not from the recombinant enzymes.

Author's Response to Decision Letter for (RSOS-200688.R0)

See Appendix A.

RSOS-200688.R1 (Revision)

Review form: Reviewer 2

Is the manuscript scientifically sound in its present form?

Yes

Are the interpretations and conclusions justified by the results?

Yes

Is the language acceptable?

Yes

Do you have any ethical concerns with this paper?

No

Have you any concerns about statistical analyses in this paper?

No

Recommendation?

Accept as is

Comments to the Author(s)

The authors have responded satisfactorily to the suggestions and made corrections accordingly.

Decision letter (RSOS-200688.R1)

Dear Dr Xu:

Title: Degradation and detoxification of azo dyes with recombinant ligninolytic enzymes from *Aspergillus* sp. with secretory overexpression in *P.pastoris*
Manuscript ID: RSOS-200688.R1

It is a pleasure to accept your manuscript in its current form for publication in Royal Society Open Science. The chemistry content of Royal Society Open Science is published in collaboration with the Royal Society of Chemistry.

On behalf of the Subject Editor Professor Anthony Stace and the Associate Editor Dr Nadia Martinez Villegas.

RSC Associate Editor:
Comments to the Author:
(There are no comments.)

RSC Subject Editor:
Comments to the Author:
(There are no comments.)

Reviewer(s)' Comments to Author:
Reviewer: 2

Comments to the Author(s)
The authors have responded satisfactorily to the suggestions and made corrections accordingly.

Appendix A

Response to Referees

Dear Editor and Reviewers,

Thank you very much for delivering your kind decision of revision of our manuscript. We are here by submitting our revised manuscript entitled “**Degradation and detoxification of azo dyes with recombinant ligninolytic enzymes from *Aspergillus sp.* with secretory overexpression in *P.pastoris***” (Manuscript ID: RSOS-200688). The manuscript has been revised in the light of your and the reviewers’ comments and suggestions. For your guidance, itemized response is appended below. All the replies are shown in blue. In the revised manuscript, all the revised and added parts have been highlighted yellow background. (Note that line and page numbering used in the replies correspond to the generated pages and line numbers by the submission system)

We hope that you as well as the reviewers will find the revised version suitable for publication. Thanks a lot again for your time and concern to our work. We are looking forward to hearing from you soon.

With my best regards,

Yours sincerely

Xiaolin Xu

The following is the point-by-point response to the Reviewers' Comments

RSC Associate Editor:

Comments to the Author:

This research contributes to a deeper understanding of the degradation mechanism of three ligninolytic enzymes. While the conclusions are supported by the data, a serious problem with the English needs to be solved. In many cases it is difficult to understand what is meant and can lead to misunderstandings that affect the validity of the presented results. Finally, it is highly recommended to place the sequences of the ligninolytic enzymes and weel as the *Aspergillus* spp. in some repository.

A: Thank you for your support and encouragement. We have revised some of the unreadable parts of the manuscript with the help of a professional language editing service. We have provided certifies of English language editing (at the end of this

document). We uploaded the sequences of the ligninolytic enzymes from *Aspergillus* sp. as part of the supporting data in submission system when we submitted the manuscript. We sincerely considering that the sequences of enzymes in repository, and we have uploaded the sequence in Dryad. (Dryad: <https://doi.org/10.5061/dryad.4tmpg4f70>). (In 3.1; page 5, line 31-32)

Thank you very much for your comments and suggestions.

Reviewer: 1

Comments to the Author(s)

The Ms deals with the degradation of synthetic dyes by cloning ligninolytic enzymes, i.e. Lac, MnP and LiP from *Aspergillus* sp. to *Pichia pastoris* in 3 plasmids. Degradation of dyes by ligninolytic enzymes is not new, and the authors mentioned some papers related to this topic. However, the Ms showed interesting results. But I have some queries to be answered:

A: Thank you for your affirmation of our work. The responses to your points are as follows:

Q1: Are the sequences of the 3 enzymes publically available in some website? Since the authors isolated the genes of the 3 ligninolytic enzymes from an *Aspergillus* spp.

A: Thanks for the reviewer's kind suggestion. The sequences of the three ligninolytic enzymes from *Aspergillus* sp. was published in one master dissertation titled with "Decolorization process of *Aspergillus* sp. TS-A and performance of recombinant decolorization enzymes" in China National Knowledge Infrastructure (CNKI, <https://www.cnki.net/>). We uploaded the sequences of the ligninolytic enzymes from *Aspergillus* sp. as part of the supporting data in submission system when we submitted the manuscript. Considering the Reviewer's suggestion, we have uploaded the sequence in Dryad. (Dryad: <https://doi.org/10.5061/dryad.4tmpg4f70>). (In 3.1; page 5, line 31-32)

Q2: The aim of the work was to degrade dyes partially or totally, means potential use

at large scale. So, how stable are the plasmids in *P. pastoris*? (How many generations? How many batches? Did the authors try to scale up the process?)

A: Many thanks to reviewer for pointing out our work. Considering the stability, the multi-copy integration, high expression, plasmid pPIC9K and *pichia pastoris* were selected as the expression system. Results verified that plasmids and recombinant enzyme activity could be detected after 5 generations, and enzyme activity could still maintain 80%-90%. It showed great stability of plasmid in recombinant yeast GS115. We will continue to modify these three ligninolytic enzymes gene to improve its activity and yield. Therefore, the following experiments can be consider using fermentation tank to produce recombinant enzymes in a large scale.

Q3: In a complex mixture of chemical pollutants (e.g. more than one toxic molecule commonly available in tannery wastewaters, how the authors are able to combine the activities of 3 enzymes?

A: Thank you for your support and encouragement to our work. The combined with three ligninolytic enzymes showed greater degradation ability than the single enzyme did. In facing of complex dye wastewater, these three enzymes could be co-immobilized with appropriate materials, such as sodium alginate, chitosan. We have also tried immobilized enzymes, and the experimental data were not shown in this article. The results showed that the immobilized enzyme could maintain the enzyme activity and recycle easily. It could be one effective method to degrade complex dye wastewater using combined enzymes.

Q4: Why the toxicity assays was performed in pepper seeds? Since Congo red is considered carcinogenic, and the result showed (decolorization, %) were from 45 to 38%? Some common mammalian cell line should be tested (e.g. normal human fibroblasts).

A: Thank you for your support and encouragement to our work. Pepper, an economic crops, is an important plant grown on a large scale in Xinjiang. In addition, Xinjiang has the largest textile industry in China. Considering potential contamination irrigation

system and water bodies by the dye effluent, this study detected the effect of dye wastewater on pepper. It showed that the length of leaf, shoot, root and weight of pepper were obviously different between dye and dye degradation products treatment. Results showed that adding degradation products could reduce the growth stress of pepper compared with adding dye solution. According to the literature, P. Aravind et al. phytotoxicity were analyzed on seeds of *Vigna radiate*. N. Asses, L. Ayed et al. used *Zea mais* and *Solanum lycopersicum* seeds to detect toxicity of dye. It can be seen that the toxicity of dyes to plants weakened after degradation. Therefore, to make it more accurate, we adjusted the “Toxicity Assay” to “Phytotoxicity Assay” (In 2.8; page 5, line 2). Once COVID-19 is over, we will try to use some common mammalian cell line cells for toxicity assay.

The decolorization rate of Congo red mentioned by reviewers was 45%-38%, which was the decolorization rate within a short period of time, while the decolorization product after 24 hours was used for the phytotoxicity assay. Through our previous experiments, the Congo red decolorization rate could reach 90% at 24 h. According to the comments of reviewers, we have added the decolorization rate of CR at 24h in the supplementary material (Figure S4) to be associated with the results of phytotoxicity assay. (Page 12, line 52)

Reference:

- [1] P. Aravind, H. Selvaraj, S. Ferro, & M. Sundaram. 2016. An integrated (electro- and bio-oxidation) approach for remediation of industrial wastewater containing azo-dyes: Understanding the degradation mechanism and toxicity assessment. *Journal of Hazardous Materials*, 318, 203–215. (10.1016/j.jhazmat.2016.07.028)
- [2] N. Asses, L. Ayed, N. Hkiri, M. Hamdi. 2018. Congo Red Decolorization and Detoxification by *Aspergillus niger*: Removal Mechanisms and Dye Degradation Pathway. *BioMed Res. Int.* 1–9. (10.1155/2018/3049686)

Figure S4. Recombinant enzymes degradation of Mordant Yellow 1 and Congo Red (a: Recombinant Lac, Lip, Mnp degradation of Mordant Yellow 1 and Congo Red; b: Combine recombinant enzymes degradation of Congo Red, Com: Lip + Mnp + Lac) Special thanks to you for your comments.

Reviewer: 2

Comments to the Author(s)

A linguistic and grammatical revision of all work is strongly recommended.

A: Thank you for your support and encouragement. We have already revised some of the linguistic and grammatical in the manuscript with the help of a professional language editing service. We have provided certifies of English language editing (at the end of this document).

Q1: The term "specific activity" is misused because activity per mg protein was not reported.

A: Thank you very much for your support and encouragement to our work. We have modified "specific activity" as "enzyme activities" in abstract. (Page 2, line 27)

Q2: The term "decolorization ratio" was used instead of "rate".

A: Thank you for correcting us. We have modified this expression throughout the text according to the comment.

Q3: The meaning of the acronym for microbiological culture media should be written.

A: The suggestion has been taken seriously. We are already added the meaning of the acronym for microbiological culture media in Materials and Methods of 2.2. Luria Bertani (LB), yeast extract peptone dextrose (YPD), minimal dextrose (MD), minimal methanol (MM), buffered glycerol complex (BMGY), buffered methanol complex media (BMMY). (Page 3, line 58-60)

Q4: According to the description of the study of pH effect on enzyme stability (2.5.1), stability is measured for 30 min and then the substrate was added. If the pH is not changed to the reference value of activity assay, activity is being measured at different pHs in each case. In this way, two studies overlap (stability/optimum pH) and should be carried out separately. Some details of the procedure are probably missing to help to understand it.

A: Thank you very much for your support and encouragement to our work. The experiment 2.5.1 was designed to explore the optimal pH of the recombinant enzymes, and it is more appropriate to describe as the optimal pH. We have changed “Effect of pH and temperature on activity and stability” to “Effect of pH and temperature on activity”. We have modified the materials and methods to explore the optimal pH for the recombinant enzymes, which is “To evaluate the optimal pH for recombinant enzymes using ABTS, MnSO₄, and veratryl alcohol as substrates, the enzymatic reaction was performed in buffers of different pH values (1, 3, 5, 7, 9)”. (Page 4, line 30-32)

Q5: Incubation time at each temperature is not reported in the thermostability study (2.5.1). It can be assumed that it is the time of the activity assay so two effects are being measured simultaneously and it is not a stability study per se.

A: Thank you very much for your support and encouragement to our work. Our experiment 2.5.1 was designed to explore the optimal temperature for the recombinant enzymes. In 2.5.1, (Page 4, line 33-34) we have modified “thermostability” as “the optimum temperature”. The statements were corrected as “The optimum temperature of the recombinant enzymes was determined by monitoring the change in activity at

different temperatures (25°C, 30°C, 35°C, 40°C, 45°C, 50°C, 60°C)". In the optimum temperature experiment, the change factor was the temperature at which the activity assay was detected.

Q6: It would be important to report how much activity is added to the reaction system (2.6) in 0.5mL of enzyme solution in each case, or how many mg of protein and the specific activity of the sample.

A: Thank you for your comments. This section have revised and modified according to the reviewer's comments. (Page 4, line 40-41) "The reaction system (3.5 mL) for decolorization contained dyes (50 mg/L) and 500µL recombinant enzymes (0.02 U). The azo dyes MY1 and CR, the anthraquinone dye disperse blue 2BLN, and the triphenylmethane dye bromophenol blue were dissolved in the buffers (pH=5). The decolorization was carried out for 5 s at 30°C."

Q7: In 2.6 the term "concentration" is used to express Decolorization but it is not explained how it is calculated. Was absorbance correlated with a standard?

A: Thank you for your careful review. We carefully checked the statement of decolorization rate in 2.6, we have re-written this part according to the reviewer's suggestion. (Page 4, line 47-51) Decolorization (%) reflected the decrease in dye concentration owing to the oxidation by recombinant enzymes. Decolorization rate was calculated by measuring the change in absorbance of the reaction mixture. The change in absorbance is not correlated with a standard.

Q8: In 2.7, what reaction volume was used for the analysis of the degradation products?

A: Thank you for your comments. We have added the reaction volume in 2.7 (Page 4, line 53-54). The reaction mixture in a total volume of 10 mL contained dyes (MY1 or CR: final concentration of 50 mg/L), buffer (50mM, pH 5) and three recombinant enzymes (0.05 U). After 24 hours, 10mL of the reaction solution was extracted with chromatographically pure ethyl acetate, and then the extracted solution was condensed to 1mL with rotary evaporator for degradation product analysis.

Q9: Why was the pepper seed bioassay selected for the toxicity study? A reference is needed for this study (2.8).

A: Pepper, an economic crops, is an important plant grown on a large scale in Xinjiang. Given that Xinjiang has the largest textile industry in China, and the potential contamination on irrigation system and water bodies by the dye effluent, this research detected the impact of dye wastewater on pepper. The growth of pepper showed that the length of leaf, shoot, root and weight of pepper were obviously different between dye and dye degradation products treatment. Results showed that adding degradation products could reduce the growth stress of pepper compared with adding dye solution. According to the literature, P. Aravind et al. phyto-toxicity were analyzed on seeds of *Vigna radiate*. N. Asses et al. the used of *Zea mais* and *Solanum lycopersicum* seeds to detect toxicity of dye. It can be seen that the toxicity of dyes to pepper weakened after degradation. Therefore, we have adjusted the “Toxicity Assay” was to “Phytotoxicity Assay” in 2.8 (Page 5, line 2). We have added reference 45 to results and discussion 3.6 (Phytotoxicity Assay; Page 12, line 52)

Reference:

[1] P. Aravind, H. Selvaraj, S. Ferro, M. Sundaram. 2016. An integrated (electro- and bio-oxidation) approach for remediation of industrial wastewater containing azo-dyes: Understanding the degradation mechanism and toxicity assessment. *Journal of Hazardous Materials*, 318, 203–215. (10.1016/j.jhazmat.2016.07.028)

[2] N. Asses, L. Ayed, N. Hkiri, M. Hamdi. 2018. Congo Red Decolorization and Detoxification by *Aspergillus niger*: Removal Mechanisms and Dye Degradation Pathway. *BioMed Res. Int.* 1–9. (10.1155/2018/3049686)

Q10: It cannot be deduced molecular mass of a protein from an SDS-PAGE unless it can be said it is monomeric based on literature or another complementary study (See in 3.2 first paragraph). How were the bands corresponding to the enzymes of interest selected from among all the bands seen in the electrophoresis in fig.1d? One could refer to the literature, for example by citing reference 33 to say that the selected band

corresponds to the laccase, but not make the deduction backwards, as was discussed. In addition, in that review several native enzymes are shown, which one is referenced in particular?

A: Thank you very much for your support and encouragement to our work. Based on the literature, most of fungal laccases are monomeric proteins (O.V. Morozova et al.). Lignin peroxidases and manganese peroxidases are monomeric heme-containing proteins (B. Sharma et al.). Considering the Reviewer's suggestion, we have re-written this part (in 3.2 first paragraph; Page 5, line 34). By citing reference to say that the selected band corresponds to the recombinant enzymes. In the introduction, we mentioned the native ligninolytic enzymes from *Aspergillus* sp. TS-A CGMCC12964, which showed excellent ability to degraded azo dyes, and we continued to research these enzymes through heterologous expression.

Reference:

[1] O.V. Morozova, G.P. Shumakovich, M.A. Gorbacheva, S.V. Shleev, A.I. Yaropolov. 2007. Blue Laccases. *Biochemistry*. 72, 1136-1150. (10.1134/S0006297907100112)

[2] B. Sharma, A.K. Dangi, P. Shukla. 2018. Contemporary enzyme based technologies for bioremediation: A review. *J. Environ. Manage.* 210, 10-22. (10.1016/j.jenvman. 2017.12.075)

Q11: In fig 1 it would be suitable to clarify what the Lac, Mnp and Lip labels correspond to. It seems appropriate to name them as described in the text: GS115-Lac, GS115-Mnp, GS115-Lip.

A: Thanks for the reviewer's kind suggestion. We have replaced Lac, Mnp, Lip with GS115-Lac, GS115-Mnp, GS115-Lip in the Fig.1. (Page 6, line 1-29)

Q12: In 3.2 line 35-36 the term recombinase is used wrongly.

A: Thank you very much for your support and encouragement to our work. We have modified "recombinase" as "recombinant enzymes" (In 3.2; Page 5, line 51-52).

Q13: In 3.3.1 line 48, references 34 and 35 are cited but in these works no thermostability of the Lip is reported.

A: Thank you for pointing out the mistake. We have added the references 17 to cited optimum temperatures of the Lip. (In 3.3.1; Page 6, line 52-53)

Q14: Lines 49-50, stability and optimum pH concepts are confused.

A: Thank you for correcting us. We carefully checked the statement of stability and optimum pH. Lines 49-50, the statements of “Differing pH stabilities were observed for the three recombinant enzymes” were revised as “The optimum pH was observed for the three recombinant enzymes”. (Page 6, line 53-54)

Q15: Line 52, pH 2.0 cannot be included in the observation because this point was not tested for the recombinant strain.

A: Thank you for correcting us. We have adjusted pH 2.0 to pH 3.0. The three recombinant enzymes showed a relatively strong activity at a broad pH range of 3.0 to 7.0. (Page 6, line 55-56)

Q16: Furthermore, it is not valid to compare the results obtained for native and recombinant enzymes since solutions with different concentrations of active enzyme were used. If possible, equal protein concentration and equal enzymatic activity per unit volume should be used.

A: Thanks for your advice. Considering the reviewer’s suggestion, the comparison between on native and recombinant enzymes with different concentrations were inappropriate. We have deleted the sentence “the recombinant Mnp remained 57.24% activity at pH 7.0, and its stability was higher than native Mnp from *Aspergillus* sp. TS-A.”

Q17: Ref 36 cannot be used to discuss 3 lignin oxidases because this is a study of MnP only.

A: Thank you for your careful review. We have re-written this part according to the reviewer’s suggestion. We have added the reference 7 to cited optimum pH values of the Lac and Lip. (Page 6, line 59)

Q18: I wonder if the resulting pH in the reaction medium is known or was measured (3.4).

A: Thank you for your comments. The resulting pH in the reaction medium was measured in 3.4. The reaction mixture contained buffer (50mM, pH=5), dyes and enzymes (details of reaction system are given in materials and methods 2.6; Page 4, line 40-42). According to the literature, bromophenol blue is an acid dye (F. B. M. Suah et al). We have added the references 38 in 3.4. (Page 7, line 44-45)

Reference:

[1] F. B. M. Suah, M. Ahmad, & M. N. Taib. 2003. Applications of artificial neural network on signal processing of optical fibre pH sensor based on bromophenol blue doped with sol-gel film. *Sensors and Actuators B: Chemical*, 90(1-3), 182–188. (10.1016/s0925-4005(03)00026-1)

Q19: In 3.4, line 42-43, MO dye decolorization reported by Jing Sun et al. was done by laccase from *G. lucidum* in presence of the redox mediator ABTS 0,025mM, not directly as mentioned.

A: Many thanks to reviewer for pointing out our work. We have carefully revised the sentence “Jing Sun et al. reported that the methyl orange (MO) was directly decolourized by the laccase obtained from *Ganoderma lucidum* about 57.48%” to “J. Sun et al. reported that methyl orange was decolorized by laccase obtained from *Ganoderma lucidum* by about 57.48% in the presence of the redox mediator ABTS” in 3.4 Results and Discussion. (Page 7, line 45-48)

Q20: It would be good to know the value of the decolorization at 24h of MY and CR to correlate it with the results of FTIR and GCMS.

A: Thank you for your comments. According to our previous experiment results, the decolorization of MY1 and CR at 24h were added to the supplementary material (**Figure S4**) to be associated with the results of FTIR and GC-MS. We have added this part in 3.5. “The recombinant enzymes were efficient in the decolorization of MY1 and CR. After 24 h, the decolorization rates of MY1 by the recombinant Lac, LiP, and MnP

were 67%, 58%, and 54%, respectively, and those of CR were 89%, 91%, and 90%, respectively (Supplementary Fig. S4). To understand the transformations of the azo dyes catalyzed by the recombinant enzymes, two analytical tools were used: FTIR and GC-MS". (Page 8, line 18-22)

Figure S4. Recombinant enzymes degradation of Mordant Yellow 1 and Congo Red (a: Recombinant Lac, LiP, MnP degradation of Mordant Yellow 1 and Congo Red; b: Combine recombinant enzymes degradation of Congo Red, Com: LiP + MnP + Lac)

Q21: The retention times of the compounds detected in the GC-MS could be reported.

A: Many thanks to reviewer for pointing out our work. We have added the retention times of the compounds detected in the Table 2 and Table 3.

Q22: Legend in figure 5 must be corrected.

A: Thank you very much for your support and encouragement to our work. We have corrected legend in figure 5. (Page 12, line 47-50)

Q23: In 3.5.3 figure 6 is mentioned but there is no figure 6 in the document.

A: Thank you for your support and encouragement. In 3.5.3, page 10, line 5, the figure 6 was corrected as figure 5.

Q24: It should be clarified what the abbreviation "com" means in Table 4.

A: Thank you very much for your support and encouragement to our work. We have added the "Combined: Lip +Mnp +Lac" in table 4.

Q25: The legend in figure S1 must be corrected, see the order of the three enzymes results.

A: Thank you for correcting us. We have corrected the mistake in figure S1 of legend in the Supplementary material.

Q26: In figure S4 the legend should be corrected too, because the result shown is not from the recombinant enzymes.

A: Thank you for your comments. We have removed the Figure S4 (Effect of pH on *Aspergillus* sp. TS-A extracellular recombinant enzymes activity) according to the modification of the article.

Thank you very much for your comments and suggestions.

Certificate of English Language Editing

Manuscript Title:

Degradation and detoxification of azo dyes with recombinant ligninolytic enzymes from *Aspergillus* sp. with secretory overexpression in *P. pastoris*

Date of Revision:

June 25, 2020

Abstract:

Ligninolytic enzymes including laccase (Lac), manganese peroxidase (MnP), and lignin peroxidase (LiP) have attracted much attention in the degradation of contaminants. Genes of Lac (1827bp), MnP (1134bp), and LiP (1119bp) were cloned from *Aspergillus* sp. TS-A, and the recombinant Lac (69 KDa), MnP (45 KDa), and LiP (35 KDa) were secretory expressed in *Pichia pastoris* GS115, with enzyme activities of 34 U/L, 135.12 U/L, 103.13 U/L, respectively. Dyes of different structures were treated via the recombinant ligninolytic enzymes under the optimal degradation conditions, and the result showed that the decolorization rate of Lac on Congo red (CR) in 5 s was 45.5%. Fourier-transform infrared (FTIR) spectroscopy, gas chromatography–mass spectrometry (GC-MS) analysis, and toxicity tests further proved that the ligninolytic enzymes could destroy the dyes, both those with one or more azo bonds, and the degradation products were nontoxic. Moreover, the combined ligninolytic enzymes could degrade CR more completely compared with the individual enzymes. Remarkably, besides azo dyes, ligninolytic enzymes could also degrade triphenylmethane and anthracene dyes. This suggests that ligninolytic enzymes from *Aspergillus* sp. TS-A have potential for application in the treatment of contaminants.

This document certifies that the manuscript listed above was copy edited for proper English language at LetPub. All of our language editors are native English speakers with long-term experience in editing scientific and technical manuscripts. We are committed to leveling the playing field for researchers whose native language is not English.

- Neither the research content nor the authors' intended meaning were altered in any way during the editing process.
- Documents receiving this certification should be considered ready for publication where language issues are concerned. *However, the authors may accept or reject LetPub's suggestions and changes at their own discretion.*
- If you have any questions or concerns about this edited document, please contact us at support@letpub.com

LetPub is an author service brand owned and operated by Accdon LLC. Headquartered in the Boston area, we are a full-spectrum author services company with a large team of US-based certified language and scientific editors, ISO 17001 accredited translators, and professional scientific illustrators and animators. We advocate ethical publication practices and are an official member of the Committee on Publication Ethics (COPE).

For more information about our company, services, and partnership programs, please visit www.letpub.com.

© 2020 Accdon, LLC. All Rights Reserved. Tel: 1-781-202-9968 Email: info@accdon.com Address: 400 Fifth Ave, Suite 530, Waltham, MA 02451, United States